# Determinants of anemia level among reproductive-age women in 29 Sub-Saharan African countries: A multilevel mixed-effects modelling with ordered logistic regression analysis

Kusse Urmale Mare[1]*, Setognal Birara Aychiluhm[2], Kebede Gemeda Sabo[1], Abay Woday Tadesse[3], Bizunesh Fentahun Kase[3], Oumer Abdulkadir Ebrahim[3], Tsion Mulat Tebeje[4], Getahun Fentaw Mulaw[5], Beminate Lemma Seifu[3]

1 Department of Nursing, College of Medicine and Health Sciences, Samara University, Samara, Ethiopia,
2 Department of Epidemiology & Biostatistics, Institute of Public Health, College of Medicine & Health Sciences, University of Gondar, Gondar, Ethiopia, 3 Department of Public Health, College of Medicine and Health Sciences, Samara University, Samara, Ethiopia, 4 School of Public Health, College of Health Sciences and Medicine, Dilla University, Dilla, Ethiopia, 5 School of Public Health, College of Medicine and Health Sciences, Woldia University, Woldia, Ethiopia

* kussesinbo@gmail.com

**Data Availability Statement:** The raw dataset used and analyzed in this study can be accessed from

## Abstract

### Background

Despite the implementation of different nutritional and non-nutritional interventions, 43% of reproductive-age women in Africa suffer from anemia. Recent evidence also shows that none of the Sub-Saharan African (SSA) countries are on the track to achieve the nutrition target of 50% anemia reduction by 2030. To date, information on the level of anemia and its determinants among reproductive-age women at the SSA level is limited. Thus, this study aimed to estimate the pooled prevalence of anemia level and its determinants in SSA countries.

### Methods

We used a pooled data of 205,627 reproductive-age women from the recent demographic and health surveys of 29 SSA countries that were conducted between 2010–2021. A multilevel mixed-effects analysis with an ordered logistic regression model was fitted to identify determinants of anemia level and the deviance value was used to select the best-fitted model. First, bivariable ordinal logistic regression analysis was done and the proportional odds assumption was checked for each explanatory variable using a Brant test. Finally, in a multivariable multilevel ordinal logistic regression model, a p-value<0.05 and AOR with the corresponding 95% CI were used to identify determinants of anemia level. All analyses were done using Stata version 17 software.

the DHS website (https://dhsprogram.com/data/dataset_admin/index.cfm).

**Funding:** The author(s) received no specific funding for this work.

**Competing interests:** The authors have declared that no competing interests exist.

**Abbreviations:** AOR, Adjusted Odds Ratio; CI, Confidence Interval; DHS, Demographic and Health Survey; ICC, Intra Class Correlation Coefficient; LMICs, Low and Middle-Income Countries; LL, Log-Likelihood; MOR, Median Odds Ratio; PCV, Proportional Change in Variance; SSA, Sub-Saharan Africa.

## Results

The pooled prevalence of anemia among women of reproductive age in SSA was 40.5% [95% CI = 40.2%-40.7%], where 24.8% [95% CI: 24.6%-25.0%], 11.1% [95% CI = 10.9%-11.2%], and 0.8% [95% CI = 0.7%-0.8%] had mild, moderate, and severe anemia, respectively. The prevalence significantly varied from the lowest of 13% in Rwanda to the highest of 62% in Mali, and anemia was found as a severe public health problem (prevalence of $\geq$ 40%) in 18 countries. The regression result revealed that polygamous marriage, women and husband illiteracy, poor household wealth, shorter birth interval, non-attendance of antenatal care, underweight, unimproved toilet and water facilities, and low community-level women literacy were positively linked with high anemia level. Additionally, the likelihood of anemia was lower in women who were overweight and used modern contraception.

## Conclusions

Overall results showed that anemia among women of reproductive age is a severe public health problem in SSA countries, affecting more than four in ten women. Thus, enhancing access to maternal health services (antenatal care and contraception) and improved sanitation facilities would supplement the existing interventions targeted to reduce anemia. Moreover, strengthening women's education and policies regulating the prohibition of polygamous marriage are important to address the operational constraints.

## Background

Anemia is the most common nutritional problem disproportionately affecting women of reproductive age in all settings due to women's physiologic state of menstruation, pregnancy, and lactation [1]. Although the risk of exposure to anemia differs across preconception, pregnancy, and lactation periods, its occurrence throughout women's reproductive life is associated with poor maternal, newborn, and child health outcomes [2–10]. Evidence has shown that maternal pre-conceptional anemia leads to spontaneous abortion [11], low birth weight and fetal growth restriction [9, 10], and long-term childhood neurodevelopmental disorders like autism, attention deficit hyperactive disorders, and intellectual disability [2].

Moreover, gestational anemia results in an increased risk of antepartum, intrapartum, and postpartum maternal hemorrhage [12–14] and subsequent blood transfusions [5, 6, 13–15], pregnancy-induced hypertension [5, 13], prolonged and induced labor [13], cesarean birth [3–6, 13], puerperal infections [15], maternal shock, longer hospitalization and critical care admission [5], and mortality [7, 8, 12, 16, 17]. Likewise, the risk of poor neonatal outcomes such as low birth weight [3, 4, 13, 16], small-for-gestational-age [3, 5, 13], preterm birth [3, 4, 6, 12, 13], low Apgar score [3, 5, 6], stillbirth [12, 13], and neonatal and perinatal mortality [5, 13, 16] is higher among neonates and children born to anemic mothers. In addition, low maternal hemoglobin level during pregnancy was also found to increase the risk of placenta previa [5] and abruption, fetal malformation, growth restriction [12], and neonatal admission [6, 14].

Globally, 30% of non-pregnant and 36% of pregnant women suffer from anemia [18]. In low and middle-income countries (LMICs), anemia affects about 32% of childbearing women and more than 50% of the countries in this region had a national anemia level of 20% to 39.9% among this age group [19]. Most importantly, its prevalence in SSA countries is 43%, with the

highest prevalence in the Western African region (51%) [20]. Previous studies at national and subnational levels across African countries have identified the influence of different socio-demographic, reproductive, nutritional, and sanitation-related factors on anemia among childbearing women [21–26].

Different nutrition-specific interventions like supplementation of iron-folic acid and vitamin A, fortification of food, and improving dietary diversity and food security through agricultural initiatives have resulted in the reduction of anemia among pregnant and non-pregnant reproductive-age women [27, 28]. Additionally, the implementation of programs like deworming, malaria control, water, sanitation, and hygiene were the strategies targeted to address non-nutritional causes of anemia among reproductive-age women [28]. Despite these efforts, during the last two decades, anemia declined by only 1% in non-pregnant and 5% in pregnant women between 2000 and 2019 [18]. Moreover, the existing evidence shows that none of the LMICs including SSA countries are on the track to achieve the nutrition target of anemia reduction by 50% in 2030, indicating that the average annual rate of decline was below the level needed to achieve the target [29]. Thus, generating up-to-date national, sub-regional, and regional estimates of anemia and identification of context-specific factors at all possible levels are important for further reduction.

To date, evidence on the level of anemia and its determinants at the SSA level is limited and most of the previous studies across this region were restricted to a single country [24, 30–33] and some African countries [20, 26, 34, 35]. However, methodologically these studies did not employ multilevel modelling to account for the hierarchical nature of the demographic and health survey (DHS) data, and data were not weighted to compensate for the non-proportional allocation of the sample size and appropriate estimation of the standard error. Additionally, they did not use the proportional odds model that accounts for the ordered nature of the anemia level. Furthermore, they did not assess the influence of some reproductive health-related factors like antenatal care, place of delivery, birth spacing, parity, and contraceptive use on anemia. Thus, taking into account the methodological gaps of the previous studies, the current study aimed to estimate the pooled prevalence of anemia level and its determinants in SSA countries using data from the DHS conducted between 2010 and 2021.

## Methods

### Data source

Pooled demographic and health survey data from 29 SSA African countries were used in this study. We took into account the survey year (surveys conducted during 2010–2021), data restrictions, and the presence of outcome variables (i.e. anemia level) in the dataset for the country inclusion. Based on this, we found that, of 43 SSA countries that have a routine DHS, four have data that was collected earlier than 2010 (Central African Republic, Sudan, Eswatini, and Sao Tome and Principe), four have data restriction (Cape Verde, Equatorial Guinea, Botswana, and Eritrea), and datasets of six countries have no observation on the outcome variable (Angola, Chad, Comoros, Kenya, Senegal, and Zambia). Finally, 29 countries were considered in the analysis (Burkina Faso, Benin, Burundi, Democratic Republic of Congo, Congo, Cote D'Ivoire, Cameroon, Ethiopia, Gabon, Ghana, Gambia, Guinea, Liberia, Lesotho, Madagascar, Mali, Mauritania, Malawi, Mozambique, Nigeria, Niger, Namibia, Rwanda, Sierra Leone, Togo, Tanzania, Uganda, South Africa, and Zimbabwe).

### Data extraction and management of missing observations

The surveys across all countries employed standardized procedures to gather data on basic socio-demographic characteristics and different health indicators. In this analysis, since the

study population was reproductive-age women (15–49 years old), we extracted women's records (IR) of each country and then appended them after managing missing observations (i.e. women with missing observations on the outcome variable were dropped/excluded from the analysis). Finally, a weighted sample of 205,627 reproductive-age women (unweighted sample = 206,026) was considered in the analysis. Details about the survey's methodology can be accessed online [36].

## Study variables

**Dependent variable.** The outcome variable for the current study was "anemia level" which was categorized into "no anemia", "mild anemia (hemoglobin level of 10.0–10.9 g/dl for pregnant women and 10.0–11.9 g/dl for nonpregnant women)", "moderate anemia (hemoglobin level 7.0–9.9 g/dl)", and "severe anemia (hemoglobin level less than 7.0 g/dl)". In DHS, anemia level was assessed using hemoglobin level adjusted for altitude and pregnancy status. In this regard, a hemoglobin level of 12 g/dL, and 11 g/dL was considered as the cut-off point to determine anemia among non-pregnant and pregnant women, respectively [37].

**Independent variables.** For analysis purposes, explanatory variables were grouped into individual and community-level variables. Individual-level variables included age, marital status, age at marriage, nature of marriage, women and partner's education, women's employment, family size, possession of bed net, sex of household head, wealth index, parity, birth interval, pregnancy status, current contraceptive use, antenatal care visit, place of delivery, delivery by cesarean section, intake of iron supplementation, body mass index, and history of pregnancy loss. While, residence, region, toilet facility, source of drinking water, distance to a health facility, community-level women's employment, and community-level women's literacy were community-level variables.

Toilet facility was grouped as "improved" and "unimproved" based on DHS guidelines. Improved toilet facilities include (a flush-to-piped sewer system, flush-to-septic tank, flush-to-pit latrine, pit latrine—ventilated improved pit, pit latrine with slab, and composting toilet). Unimproved toilet facilities include (flush-to-somewhere else, pit latrine—without a slab or open pit, bucket toilet, hanging toilet/latrine, and other DHS categories) [38].

Source of drinking water was grouped as "improved" and "unimproved" based on DHS guidelines. Water sources including household connections, public standpipes, boreholes, protected dug wells, protected springs, and rainwater collection were grouped as improved water sources. While unimproved water sources include unprotected wells, unprotected springs, surface water (from river, dam, or lake), vendor-provided water, bottled water (unless water for other uses is available from an improved source), and tanker truck-provided water [38].

Community-level variables (i.e. community-level women's employment and community-level women's literacy) were generated by aggregating the individual-level observations at the cluster level and median values were used to categorize the generated variables as "low" and "high".

**Data management and statistical analysis.** The datasets of all SSA countries were accessed from the DHS program's official database after securing the authorization letter. Then the dataset of each country was cleaned for missing and inconsistent observations. Finally, the cleaned datasets of 29 countries were appended into a single dataset and variable extraction was done based on the literature. Stata software version 17 was used for data analysis and data were weighted to compensate for the non-representativeness of the sample and obtain reliable estimates. Frequency and percentages were used to present descriptive results. To account for the hierarchal nature of DHS data and the ordinal nature of the outcome variable (i.e. anemia level), a multilevel mixed-effect ordered logistic regression model was fitted

**Table 1. Generalized variance inflation factor values for the predictors of anemia level among reproductive-age women in 29 SSA countries.**

| Predictor variables | VIF | Tolerance (1/VIF) | GVIF |
|---|---|---|---|
| Woman's education | 1.76 | 0.57 | 1.02 |
| Partner's education | 1.65 | 0.61 | 1.01 |
| Nature of marriage | 1.22 | 0.82 | 1.00 |
| Household wealth | 1.45 | 0.69 | 1.01 |
| Family size | 1.05 | 0.95 | 1.00 |
| Birth interval | 1.01 | 0.99 | 1.00 |
| Currently pregnant | 1.08 | 0.93 | 1.00 |
| Contraceptive use | 1.21 | 0.83 | 1.01 |
| Antenatal care visit | 1.41 | 0.71 | 1.01 |
| Place of delivery | 1.34 | 0.74 | 1.01 |
| Body mass index | 1.09 | 0.92 | 1.01 |
| Pregnancy loss | 1.01 | 0.99 | 1.00 |
| SSA regions | 1.11 | 0.90 | 1.00 |
| Toilet facility | 1.28 | 0.78 | 1.01 |
| Water source | 1.16 | 0.86 | 1.00 |
| Community-level women employment | 1.00 | 0.99 | 1.00 |
| Community-level women literacy | 1.03 | 0.97 | 1.00 |

GVIF = Generalized variance inflation factor; VIF = Varaiance inflation factor.

to identify determinants of anemia level. First, unadjusted ordinal logistic regression was done and likelihood ratio and Brant tests was conducted for each predictor variable as post-estimation [39, 40]. Therefore, in the unadjusted analysis, explanatory variables that satisfied the proportional odds assumption and had a p-value less than 0.25 were selected for inclusion in the final multivariable regression model [41, 42]. A collinearity diagnostic was carried out using a generalized variance inflation factor (GVIF) due to the categorical nature of the outcome variable, the complexity of the sampling design used, and the hierarchical structure of the fitted model. Thus, to get GVIF for the explanatory variables, the variance inflation factor (VIF) was first obtained as a post-estimation result and then GVIF was computed using a formula, $GVIF = VIF^{[1/(2*df)]}$, where df (degree of freedom) = 16). As presented in Table 1, GVIF values for all predictor variables included in the final model were less than five indicating that there was no multicollinearity (Table 1).

## Model building and selection

Four hierarchal models were fitted to select the model that best fits the data: a model without explanatory variables to test random variability in the intercept (model I), a model with only individual-level explanatory variables (model II), a model with only community-level variables (model III), and a model with both individual and community-level variables (model IV). Deviance (i.e. -2*LL) was used for model selection and the model with the lowest deviance value was considered as a best-fitted model [43]. Finally, in the multivariable multilevel ordinal logistic regression model, a p-value less than 0.05 and an adjusted odds ratio with the corresponding 95% confidence interval were used to declare the statistical significance of the independent variables. In addition, the random variability in the prevalence of anemia across clusters was estimated by intra-class correlation coefficient (ICC), proportion change in variance (PCV), and median odds ratio (MOR).

### Ethical approval

Permission to access the data (AuthLetter_179743) used in the present study was granted from a measure demographic and health survey via an online request at http://www.dhsprogram. com. The accessed data were only used for this registered study and are publicly available from the program's official database.

## Results

### Sociodemographic characteristics

Of 205,627 reproductive-age women included in the study, 80,499 (39%) were between the ages of 15 and 24 years. The majority (63.9%) of women were married and more than three-fourths (76.9%) were in a monogamous relationship. Less than one-third (30%) of women had no formal education, while 75,234 (36.7%) attended higher education. Moreover, 127,284 (62%) women resided in rural areas and 41,978 (20%) were from the Eastern African region. It was also found that 107,413 (52%) and 61,088 (29.7%) women lived in households with unimproved toilet and water facilities, respectively (Table 2).

### Obstetrics and reproductive characteristics

Less than half (46.7%) of women included in the analysis were married before the age of 18 years and 121,267 (59.0%) were multiparous. In addition, 185,891 (90.4%) women had a birth interval of 24 months or more, and 52,652 (25.6%) were using contraceptives. The majority (77.8%)of women consumed iron-folic acid supplementation, 53,050 (25.8%) were overweight, and 28,709 (14.0) had a history of pregnancy loss (Table 3).

### Prevalence of anemia among reproductive-age women in SSA countries

The pooled prevalence of anemia among reproductive-age women in SSA countries was 40.5% (95% CI = 40.2%-40.7%), with mild, moderate, and severe anemia accounting for 24.8% (95% CI = 24.6%-25.0%), 11.08% (95% CI = 10.9%-11.2%), 0.8% (95% CI = 0.75%-08%), respectively. The prevalence greatly varied across the countries, with Mali having the highest prevalence (63%) and Rwanda having the lowest (13%). Our result also showed that anemia was a severe public health problem (prevalence of $\geq 40$) in 18 of the countries included in the analysis and was a moderate problem in the remaining countries except for Rwanda. Furthermore, sub-group analysis revealed that Western (51%) and Southern (37%) African regions had a higher anemia prevalence compared to other regions (Fig 1).

### Random effect analysis

The value of ICC in the null model indicates that there was a 17.9% variation in the level of anemia among reproductive-age women across the clusters and the variability was reduced to about 11% after considering both individual and community-level variables in the full model. The PCV value of the final model showed that the combined effect of individual and community-level variables accounted for 44.9% of the variation in the prevalence of anemia at the community level. Additionally, the presence of heterogeneity in the anemia level between clusters was indicated by the MOR with a value of 2.19 in the null model. This shows that the likelihood of anemia among women in the clusters with higher anemia prevalence was more than two-fold compared to those in the clusters with a lower anemia level. Model IV had the lowest deviance value (i.e. 182,250) and was hence selected as the best-fitted model (Table 4).

**Table 2. Socio-demographic characteristics of reproductive-age women in 29 SSA countries (n = 205,627).**

| Characteristics | Level of anemia [n (%)] | | | | Total |
|---|---|---|---|---|---|
| | No anemia | Mild | Moderate | Severe | |
| **Age** | | | | | |
| 15–24 | 46,269 (38.8) | 21,825 (39.9) | 11,483 (39.1) | 922 (38.9) | 80,499 (39.1) |
| 25–34 | 38,431 (32.3) | 16,858 (30.8) | 9,294 (31.6) | 742 (31.3) | 65,326 (31.8) |
| 35–49 | 34,445 (28.9) | 16,050 (29.3) | 8,598 (29.3) | 709 (29.9) | 59,802 (29.1) |
| **Marital status** | | | | | |
| Never married | 33,405 (28.0) | 14,525 (26.5) | 7,050 (24.0) | 577 (24.3) | 55,558 (27.0) |
| Currently married | 74,919 (62.9) | 35,050 (64.0) | 19,857 (67.6) | 1,590 (67.0) | 131,416 (63.9) |
| Formerly married | 10,821 (9.1) | 5,158 (9.2) | 2,467 (8.4) | 206 (8.7) | 18,652 (9.1) |
| **Nature of marriage** | | | | | |
| Monogamy | 59,382 (79.3) | 26,359 (75.2) | 14,236 (71.7) | 1,103 (69.4) | 101,080 (76.9) |
| Polygamy | 15,511 (20.7) | 8,682 (24.8) | 5,620 (28.3) | 487 (30.6) | 30,301 (23.1) |
| **Woman's education** | | | | | |
| No formal education | 32,508 (27.3) | 17,780 (32.5) | 11,772 (40.1) | 1,088 (45.8) | 63,148 (30.7) |
| Primary education | 40,679 (34.2) | 17,616 (32.2) | 8,051 (27.4) | 690 (29.1) | 67,055 (32.6) |
| Higher education | 45,939 (38.6) | 19,337 (35.3) | 9,552 (32.5) | 596 (25.1) | 75,234 (36.7) |
| **Woman's employment** | | | | | |
| Not working | 49,475 (41.5) | 22,819 (41.7) | 12,568 (42.8) | 1,212 (51.1) | 86,074 (41.9) |
| Working | 69,626 (58.5) | 31,883 (58.3) | 16,798 (57.2) | 1,161 (48.9) | 119,469 (58.1) |
| **Partner Education** | | | | | |
| No formal education | 24,949 (32.0) | 14,091 (37.8) | 9,512 (46.3) | 902 (54.3) | 49.455 (36.0) |
| Primary education | 24,676 (31.6) | 10,538 (28.3) | 4,679 (22.9) | 389 (23.4) | 40,30 (29.3) |
| Higher education | 28,390 (36.9) | 12,534 (33.9) | 6,349 (30.8) | 371 (22.3) | 47,744 (34.7) |
| **Head of household** | | | | | |
| Male | 85,086 (71.4) | 39,712 (72.6) | 21,723 (73.9) | 1,689 (71.2) | 148,211 (72.1) |
| Female | 34,059 (28.6) | 15,021 (27.4) | 7,652 (26.1) | 648 (28.8) | 57,416 (27.9) |
| **Household wealth** | | | | | |
| Poor | 55,206 (46.3) | 23,427 (42.8) | 11,434 (38.9) | 802 (33.8) | 90,867 (44.2) |
| Middle | 23,053 (19.4) | 10,864 (19.9) | 5,861 (19.9) | 424 (17.9) | 40,203 (19.6) |
| Rich | 40,886 (34.3) | 20,442 (37.3) | 12,080 (41.1) | 1,148 (48.3) | 74,555 (26.2) |
| **Family size** | | | | | |
| 1–5 | 57,745 (48.5) | 24,543 (44.8) | 12,810 (43.6) | 1,066 (44.9) | 96,164 (46.8) |
| > 5 | 61,400 (51.5) | 30,190 (55.2) | 16,565 (56.4) | 1,308 (55.1) | 109,463 (53.2) |
| **Possession of bed net** | | | | | |
| No | 31,048 (29.9) | 14,733 (29.0) | 7,869 (27.9) | 654 (29.5) | 54,304 (29.3) |
| Yes | 72,905 (70.1) | 36,097 (71.0) | 20,339 (72.1) | 1,566 (70.5) | 130,907 (70.7) |
| **Residence** | | | | | |
| Urban | 44,904 (37.7) | 21,418 (39.1) | 11,246 (38.3) | 774 (32.6) | 78,343 (38.1) |
| Rural | 74,241 (62.3) | 33,315 (60.9) | 18,128 (61,7) | 1,600 (67.4) | 127,284 (61.9) |
| **SSA regions** | | | | | |
| Central Africa | 19,303 (16.2) | 7,904 (14.4) | 2,432 (8.3) | 112 (4.7) | 29,751 (14.5) |
| Eastern Africa | 28,751 (24.1) | 9,418 (17.2) | 3,493 (11.9) | 316 (13.3) | 41,978 (20.4) |
| Southern Africa | 25,075 (21.1) | 10,775 (19.7) | 3,476 (11.8) | 339 (14.3) | 39,665 (19.3) |
| Western Africa | 46,015 (38.6) | 26,636 (48.7) | 19,974 (68.0) | 1,607 (67.7) | 94,233 (45.8) |
| **Distance to health facility** | | | | | |
| Big problem | 44,114 (37.0) | 21,604 (39.5) | 11,608 (39.5) | 1,032 (43.5) | 78,359 (38.1) |
| Not a big problem | 75,022 (63.0) | 33,124 (60.5) | 17,761 (60.5) | 1,342 (56.5) | 127,249 (61.9) |

(*Continued*)

**Table 2.** (Continued)

| Characteristics | Level of anemia [n (%)] | | | | Total |
|---|---|---|---|---|---|
| | **No anemia** | **Mild** | **Moderate** | **Severe** | |
| **Toilet facility** | | | | | |
| Improved | 58,996 (49.5) | 25,355 (46.3) | 12,894 (43.9) | 952 (40.1) | 98,198 (47.8) |
| Unimproved | 60,142 (50.5) | 29,372 (53.7) | 16,477 (56.1) | 1,412 (59.9) | 107,413 (52.2) |
| **Water source** | | | | | |
| Improved | 84,945 (71.3) | 38,116 (69.6) | 19,941 (67.9) | 1,523 (64.2) | 144,525 (70.3) |
| Unimproved | 34,191 (28.7) | 16,614 (30.4) | 9,433 (32.1) | 851 (35.8) | 61,088 (29.7) |
| **Community-level women employment** | | | | | |
| Low | 56,697 (47.6) | 26,460 (48.3) | 14,130 (48.1) | 1,105 (46.6) | 98.393 (47.9) |
| High | 62,447 (52.4) | 28,273 (51.7) | 15,244 (51.9) | 1,269 (53.4) | 107,234 (52.1) |
| **Community-level women literacy** | | | | | |
| High | 24,919 (20.9) | 11,690 (21.4) | 6,745 (23.0) | 525 (22.1) | 43,880 (21.3) |
| Low | 94,226 (79.1) | 43,043 (78.6) | 22,629 (77.0) | 1,848 (77.9) | 161,747 (78.7) |

## Determinants of anemia level among reproductive-age women in SSA countries

Only explanatory variables that satisfied the proportional odds assumptions were included in the final multilevel ordinal logistic regression model. It was revealed that women in a polygamous union [AOR (95% CI) = 1.16(1.12, 1.21)] had a 16% greater chance of having a higher anemia versus the combination of no anemia, mild anemia, and moderate anemia levels when compared with those in a monogamous union. For women with no formal education [AOR (95% CI) = 1.36(1.30, 1.42)] and those whose husbands did not attend formal school [AOR (95% CI) = 1.11(1.06, 1.17)], the likelihood of having a high anemia level was 1.36 and 1.11 times higher, respectively. Furthermore, compared to women from wealthy families, women from the middle [AOR (95% CI) = 1.10(1.02, 1.12)] and low-income [AOR (95% CI) = 1.12 (1.07, 1.17)] households had a greater risk of experiencing higher levels of anemia.

In addition, our analysis revealed that the odds of higher anemia level were greater for women with a shorter birth interval [AOR (95% CI) = 1.27(1.22, 1.32)], those who did not receive antenatal care [AOR (95% CI) = 1.30(1.23, 1.38)] and currently pregnant [AOR (95% CI) = 1.36 (1.29, 1.43)]. Likewise, holding the effect of other covariates constant, women who were underweight [AOR (95% CI) = 1.21(1.10, 1.27)], had unimproved toilet [AOR (95% CI) = 1.20 (1.16, 1.24)], unimproved water source [AOR (95% CI) = 1.34(1.30, 1.38)], and lived in the community with low-level women literacy [AOR (95% CI) = 1.21(1.14, 1.29)] were more likely to suffer from higher anemia level compared to their counter groups. On the contrary, overweight [AOR (95% CI) = 0.85(0.81, 0.89)] and contraceptive use [AOR (95% CI) = 0.57 (0.55, 0.59)] were associated with decreased odds of anemia (Table 5).

## Discussion

Due to the physiologic nature of menstruation, pregnancy, and lactation, women of reproductive age are at greater risk of anemia. This study aimed to identify determinants of anemia levels among women of reproductive age in SSA countries. For instance, it was found that the pooled prevalence of anemia among reproductive-age women in SSA countries was 40.5% (95% CI = 40.2%-40.7%), which was almost consistent with the previous study in East Africa that reported the prevalence ranging from 40.7% to 42.9% [23]. However, our finding was

**Table 3. Obstetric and reproductive characteristics of reproductive-age women in 29 SSA countries (n = 205,627).**

| Characteristics | Level of anemia [n (%)] | | | | Total |
|---|---|---|---|---|---|
| | **No anemia** | **Mild** | **Moderate** | **Severe** | |
| **Age at marriage** | | | | | |
| ≥ 18 year | 46,780 (54.6) | 20,946 (52.1) | 11,309 (50.7) | 881 (49.1) | 79,917 (53.3) |
| < 18 year | 38,960 (45.4) | 19,262 (47.9) | 11,015 (49.3) | 915 (50.9) | 70,151 (46.7) |
| **Parity** | | | | | |
| Nulliparous | 32,250 (27.1) | 14,393 (26.3) | 7,489 (22.5) | 697 (29.4) | 54,829 (26.7) |
| Primiparous | 17,241 (14.3) | 7,829 (14.3) | 4,152 (14.1) | 308 (13.0) | 29,531 (14.3) |
| Multiparous | 69,654 (58.4) | 32,511 (59.4) | 17,734 (60.4) | 1,368 (57.6) | 121,267 (59.0) |
| **Birth interval** | | | | | |
| ≥ 24 months | 108,134 (90.8) | 49,484 (90.4) | 26,158 (89.1) | 2,115 (89.1) | 185,891 (90.4) |
| < 24 months | 11,011 (9.2) | 5,249 (9.6) | 3,217 (10.9) | 259 (10.9) | 19,735 (9.6) |
| **Currently pregnant** | | | | | |
| No | 110,132 (92.4) | 50.621 (92.5) | 24,755 (84.3) | 2,076 (87.5) | 187,585 (91.2) |
| Yes | 9,012 (7.6) | 4,112 (7.5) | 4,620 (15.7) | 297 (12.5) | 18,042 (8.8) |
| **Contraceptive use** | | | | | |
| Not using | 83,734 (70.3) | 42,436 (77.5) | 24,714 (84.1) | 2,091 (88.1) | 152,975 (74.4) |
| Using | 35,411 (29.7) | 12,297 (22.5) | 4,661 (15.9) | 283 (11.9) | 52,652 (25.6) |
| **Antenatal care visit** | | | | | |
| Yes | 56,432 (91.1) | 26,818 (90.4) | 14,052 (86.2) | 930 (78.9) | 98.232 (90.1) |
| No | 5,510 (8.9) | 2,836.6 (9.6) | 2,252 (13.8) | 249 (21.3) | 10,847 (9.9) |
| **Place of delivery** | | | | | |
| Health facility | 44,333 (71.6) | 20,483 (69.1) | 10,384 (63.7) | 614 (52.1) | 75,815 (69.5) |
| Home | 17,598 (28.4) | 9,165 (30.9) | 5,923 (36.3) | 565 (47.9) | 33,250 (30.5) |
| **Delivery by CS** | | | | | |
| No | 58,133 (94.0) | 28,062 (94.7) | 15,539 (95.5) | 1,132 (96.1) | 102,866 (94.5) |
| Yes | 3,709 (6.0) | 1,560 (5.3) | 730 (4.5) | 45 (3.8) | 6,044 (5.5) |
| **Took iron-folic acid** | | | | | |
| Yes | 48,364 (78.1) | 23,178 (78.2) | 12,469 (76.5) | 824 (69.9) | 84,835 (77.8) |
| No | 13,567 (21.9) | 6,473 (21.8) | 3,832 (23,5) | 355 (30.1) | 24,227 (22.2) |
| **Body mass index** | | | | | |
| Normal weight | 74,526 (62.5) | 35672 (65.2) | 19,203 (65.4) | 1,535 (64.7) | 130.937 (63.7) |
| Underweight | 11,879 (10.0) | 5,980 (10.9) | 3,372 (11.5) | 407 (17.1) | 21,640 (10.5) |
| Overweight | 32,739 (27.5) | 11,081 (23.9) | 6,799 (23.1) | 431 (18.2) | 53,050 (25.8) |
| **Pregnancy loss** | | | | | |
| No | 103,029 (86.5) | 46.662 (85.3) | 25,183 (85.7) | 2,034 (85.6) | 176,908 (86.0) |
| Yes | 16,109 (13.5) | 8,068 (14.7) | 4,192 (14.3) | 340 (14.3) | 28,709 (14.0) |

higher than the studies in 21 SSA countries (36.9%) [34], Ethiopia (37.5%) [21], Uganda (32%) [32], Rwanda (19%) [31], and Pakistan (18%) [22]. This discrepancy could be partly explained by differences in the scope of the study and population. For instance, previous studies were carried out at a single-country level [21, 31] and limited to some African countries [34]. Furthermore, these studies were either conducted only on pregnant women [21] or non-pregnant women [22, 31] which differ from the population of the present study (i.e. reproductive-age women regardless of their pregnancy status).

On the contrary, the prevalence of anemia found in this study was greatly lower than the studies in Tanzania (83.5%) [33], Burkina Faso (55%) [44], Asia (53%) [45], Afghanistan (52%) [46], Gambia (44%) [24], and 27 SSA countries (43%) [20]. Likewise, our finding was

| Country | Sample size | Anemic | ES (95% CI) | Weight (%) |
|---|---|---|---|---|
| Burkina Faso | 8142 | 3912 | 48.05 (46.96, 49.13) | 3.53 |
| Benin | 7820 | 4508 | 57.65 (56.55, 58.74) | 3.47 |
| Burundi | 8464 | 3141 | 37.11 (36.08, 38.14) | 3.92 |
| DR Congo | 9151 | 3701 | 40.44 (39.44, 41.45) | 4.11 |
| Congo | 5137 | 2720 | 52.95 (51.58, 54.31) | 2.23 |
| Cote Divoire | 4542 | 2393 | 52.69 (51.23, 54.14) | 1.97 |
| Cameroon | 6347 | 2491 | 39.25 (38.05, 40.45) | 2.88 |
| Ethiopia | 14143 | 3858 | 27.28 (26.54, 28.01) | 7.71 |
| Gabon | 5125 | 2981 | 58.17 (56.82, 59.52) | 2.28 |
| Ghana | 4582 | 1934 | 42.21 (40.78, 43.64) | 2.03 |
| Gambia | 5699 | 2782 | 48.82 (47.52, 50.11) | 2.47 |
| Guinea | 5157 | 2314 | 44.87 (43.51, 46.23) | 2.26 |
| Liberia | 3971 | 1880 | 47.34 (45.79, 48.90) | 1.72 |
| Lesotho | 3155 | 820 | 25.99 (24.46, 27.52) | 1.77 |
| Madagascar | 9350 | 2626 | 28.09 (27.17, 29.00) | 5.01 |
| Mali | 5036 | 3165 | 62.85 (61.51, 64.18) | 2.33 |
| Mauritania | 7060 | 3857 | 54.63 (53.47, 55.79) | 3.08 |
| Malawi | 7876 | 2652 | 33.67 (32.63, 34.72) | 3.82 |
| Mozambique | 13003 | 6810 | 52.37 (51.51, 53.23) | 5.64 |
| Nigeria | 14503 | 8412 | 58.00 (57.20, 58.81) | 6.44 |
| Niger | 4817 | 2144 | 44.51 (43.11, 45.91) | 2.11 |
| Namibia | 4177 | 868 | 20.78 (19.55, 22.01) | 2.75 |
| Rwanda | 7167 | 932 | 13.00 (12.23, 13.78) | 6.86 |
| Sierra Leone | 7096 | 3338 | 47.04 (45.88, 48.20) | 3.08 |
| Togo | 4676 | 2156 | 46.11 (44.68, 47.54) | 2.04 |
| Tanzania | 12361 | 5709 | 46.19 (45.31, 47.06) | 5.38 |
| Uganda | 5789 | 1861 | 32.15 (30.94, 33.35) | 2.87 |
| South Africa | 2865 | 907 | 31.66 (29.95, 33.36) | 1.43 |
| Zimbabwe | 8815 | 2415 | 27.40 (26.47, 28.33) | 4.80 |
| Pooled prevalence of anemia | | | 40.45 (40.24, 40.65) | 100.00 |

-64.2   0   64.2

**Fig 1. Country-level and pooled prevalence of anemia among married reproductive-age women in 29 SSA countries.**

also lower than other previous studies [26, 30, 35, 47]. Variations in the study settings and period, sample size, and population characteristics might have resulted in these discrepancies.

The result of the multilevel ordinal logistic regression model revealed that women whose husbands had another wife were at a greater odds of being affected by a higher anemia level

**Table 4. The result of random-effect logit models in predicting the level of anemia among reproductive-age women in 29 SSA countries.**

| Parameters | Null Model | Model-II | Mode-III | Model-IV |
|---|---|---|---|---|
| Variance | 0.717 | 0.510 | 0.482 | 0.395 |
| Intraclass correlation coefficient | 0.179 | 0.134 | 0.128 | 0.107 |
| Proportion change in variance | Reference | 0.289 | 0.328 | 0.449 |
| Median odds ratio | 2.193 | 1.849 | 1.798 | 1.628 |
| **Model fitness** | | | | |
| AIC | 408,568 | 183,320 | 400,272 | 181,320 |
| BIC | 408,609 | 183,518 | 400,385 | 181,584 |
| Log-likelihood | -204,684 | -92,010 | -200,699 | -91,125 |
| Deviance | 409,368 | 184,020 | 401,398 | 182,250 |

**Table 5. A multivariable multilevel ordered logistic regression model describing factors associated with anemia level among reproductive-age women in 29 SSA countries.**

| Covariates | Adjusted analysis (AOR) | | | P-value |
|---|---|---|---|---|
| | Model 2 | Model 3 | Model 4 | |
| **Individual-level determinants** | | | | |
| **Nature of marriage** | | | | < 0.001 |
| Monogamy | 1.00 | - | 1.00 | |
| Polygamy | 1.37 (1.33. 1.42) | | 1.16 (1.12, 1.21)* | |
| **Woman's education** | | | | |
| Higher education | 1.00 | | 1.00 | |
| Primary education | 1.03 (0.99, 1.06) | - | 0.96 (1.10, 1.22) | 0.124 |
| No formal education | 1.56 (1.52, 1.62) | | 1.36 (1.30, 1.42)* | < 0.001 |
| **Partner's education** | | | | |
| Higher education | 1.00 | | 1.00 | |
| Primary education | 0.99 (0.93, 1.05) | - | 0. 94(0.89, 1.00) | 0.071 |
| No formal education | 1.51 (1.46, 1.57) | | 1.11 (1.06, 1.17)* | < 0.001 |
| **Household wealth** | | | | |
| Rich | 1.00 | | 1.00 | |
| Middle | 1.18 (1.14, 1.22) | - | 1.10 (1.02, 1.12)* | 0.032 |
| Poor | 1.33 (1.28, 1.38) | | 1.12 (1.07, 1.17)* | < 0.001 |
| **Family size** | | | | |
| 1–5 | 1.00 | - | 1.00 | |
| > 5 | 1.17 (1.14, 1.20) | | 1.00 (0.96, 1.04) | 0.206 |
| **Birth interval** | | | | |
| ≥ 24 months | 1.00 | - | 1.00 | |
| < 24 months | 1.32 (1.28, 1.36) | | 1.27 (1.22, 1.32)* | 0.008 |
| **Currently pregnant** | | | | |
| No | 1.00 | - | 1.00 | |
| Yes | 1.65 (1.58, 1.72) | | 1.36 (1.29, 1.43)* | < 0.001 |
| **Contraceptive use** | | | | |
| Not using | 1.00 | - | 1.00 | |
| Using | 0.57 (0.56, 0.61) | | 0.57 (0.55, 0.59)* | < 0.001 |
| **Antenatal care visit** | | | | |
| Yes | 1.00 | - | 1.00 | |
| No | 1.42 (1.33, 1.51) | | 1.30 (1.23, 1.38)* | 0.004 |
| **Place of delivery** | | | | |
| Health facility | 1.00 | - | 1.00 | |
| Home | 1.31 (1.26, 1.36) | | 1.10 (0.96, 1.05) | 0.948 |
| **Body mass index** | | | | |
| Normal weight | 1.00 | - | 1.00 | |
| Underweight | 1.20 (1.07, 1.15) | | 1.21 (1.10, 1.27)* | 0.001 |
| Overweight | 0.80 (0.78, 0.82) | | 0.85 (0.81, 0.89)* | < 0.001 |
| **Pregnancy loss** | | | | |
| No | 1.00 | - | 1.00 | |
| Yes | 1.18 (1.15, 1.22) | | 1.17 (1.13, 1.23)* | 0.001 |
| **Community-level determinants** | | | | |
| **SSA regions** | | | | |
| Central Africa | | 1.00 | 1.00 | |
| Eastern Africa | - | 0.88 (0.83, 0.93) | 0.83 (0.78, 0.90)* | < 0.001 |

(*Continued*)

**Table 5.** (Continued)

| Covariates | Adjusted analysis (AOR) | | | P-value |
|---|---|---|---|---|
| | Model 2 | Model 3 | Model 4 | |
| **Individual-level determinants** | | | | |
| Southern Africa | | 1.07 (1.02, 1.14) | 1.05 (0.98, 1.13) | 0.081 |
| Western Africa | | 2.14 (2.04, 2.24) | 2.12 (2.02, 2.22)* | < 0.001 |
| **Toilet facility** | | | | |
| Improved | - | 1.00 | 1.00 | |
| Unimproved | | 1.21 (1.18, 1.25) | 1.20 (1.16, 1.24)* | < 0.001 |
| **Water source** | | | | |
| Improved | - | 1.00 | 1.00 | |
| Unimproved | | 1.43 (1.39, 1.47) | 1.34 (1.30, 1.38)* | < 0.001 |
| **Community-level women employment** | | | | |
| Low | | 1.00 | 1.00 | |
| High | | 1.06 (1.04, 1.12) | 1.00 (0.96, 1.04) | 0.072 |
| **Community-level women literacy** | - | | | |
| High | | 1.00 | 1.00 | |
| Low | | 1.33 (1.26, 1.42) | 1.21 (1.14, 1.29)* | < 0.001 |

*Statistically significant variables at P-value less than 0.05.

than those in a monogamous marital structure. The possible justification for this finding is that women in a polygamous marriage have a higher risk of experiencing mental health (anxiety and depression) and psychological problems [48, 49] which have a dose-response relationship with the occurrence of anemia related to poor dietary habits [50]. Access to food and decision-making power might have also resulted in this finding.

This study found that women who did not attend formal education and those from the areas with low community-level women literacy had increased odds of high anemia level than their counter groups. This finding was consistent with the result of the previous studies [20, 45]. Likewise, it was supported by the reciprocal association between anemia and higher educational attainment reported in the other previous studies [30, 35, 51–53]. This might be because educated women have good nutritional knowledge and dietary habits and therefore less likely to suffer from nutritional deficiencies compared to uneducated women [54]. In addition, the effect of education on women's empowerment [55] which positively affects their reproductive health patterns might also explain this finding [56].

Our analysis also revealed that women whose husbands did not have formal education were more likely to have high anemia levels than those with educated husbands. This was in line with the finding of the study in Africa that reported lower odds of anemia in women of husbands with primary education [26]. The possible reason for this finding is that educated husbands have a better understanding and attitude toward maternal healthcare services and get more involved in their spousal utilization of these services [57].

According to this study, the likelihood of suffering from anemia was higher among women in households with middle and poor wealth indexes compared to those from rich households. Consistent with this finding, previous studies reported decreased odds of anemia with an improving household wealth index [20, 26, 31, 32, 45, 51–53, 58, 59]. This might be because women in households with high socioeconomic status are economically empowered and tend to have better dietary practices [60] and access to health services [61]. The effect of household wealth on food insecurity which directly affects nutritional status [62] might also be attributable to this finding.

Suboptimal birth spacing was also found to have a significant influence on maternal anemia. In this regard, women with a duration of birth interval less than 24 months had 27% increased odds of being affected by higher level anemia compared to those with optimal birth interval. This finding was in agreement with the result of the previous studies [22, 34, 46, 53, 63].

Consistent with the findings of the previous studies [21, 47, 53], the present study showed that women who did not receive antenatal care were more likely to suffer from anemia compared to those who had antenatal care attendance during the pregnancy of their index child. This is because women who did not attend antenatal clinics miss the opportunity to get preventive health services like iron supplementation, nutritional education, malaria prophylaxis, and other related interventions that have a significant effect on anemia.

Moreover, this study found that women who were pregnant during the survey had 36% higher odds of being affected by high anemia levels compared to non-pregnant women. Similarly, previous studies in Ethiopia [25, 30] and Gambia [24] reported a higher risk of anemia among pregnant women. This might be related to an increased nutritional demand and physiologic changes during pregnancy.

It was also found that women with undernutrition had a 21% higher likelihood of being affected by anemia than those with normal body mass index. This finding aligns with the previous studies that showed a positive association between anemia and lower body mass index [20, 31, 45]. Additionally, lower anemia risk among women with normal body mass index reported in the other study also supported our finding [30, 53].

Furthermore, this study highlighted the effect of sanitation status on anemia occurrence among women. Accordingly, women from households with unimproved toilets and water facilities were found to have a higher anemia risk compared to their counterparts. Our result was in agreement with the studies in which water source [20, 31, 32, 45, 52, 64] and toilet facility [23, 26, 45, 53] were reported to have a significant influence on anemia.

In this study, maternal contraceptive use was found as a protective factor against anemia. For instance, women who were using contraceptives had about 43% reduced odds of having anemia than non-users. Similarly, previous studies in African settings consistently reported a decreased likelihood of anemia among contraceptive users [24–26, 31, 51–53]. This association might be partly explained by the effect of iron-containing contraceptive pills on increasing hemoglobin concentration [65, 66] and decreasing menstrual blood loss [67].

## Strengths and limitations

A multi-country analysis of data from DHS data of 29 SSA countries, consistent use of Hemo-Cue blood hemoglobin testing system for measuring hemoglobin level in all surveys, use of a larger sample size, and an advanced statistical method (i.e. multilevel ordinal logistic regression model), which takes into account the hierarchical structure of the data and ordered nature of the outcome variable are the main strengths of this study. However, it is impossible to show the cause-and-effect relationship between the explanatory and outcome variables due to the cross-sectional nature of the data. The only tests employed to assess the proportional odds assumption were the Brant and likelihood ratio tests. Furthermore, since the study participants were asked about the events that took place five years or more preceding the survey, there might be a recall bias.

## Conclusions

The overall result showed that anemia among women of reproductive age is a severe public health problem in SSA countries, affecting more than four in ten women. Nature of marriage,

women and husband education, household wealth, birth interval, contraceptive use, antenatal care, nutritional status, and sanitation (toilet and water) facilities were significantly associated with anemia. Thus, enhancing access to maternal health services (antenatal care and contraception) and improved sanitation facilities would supplement the existing nutritional interventions targeted to reduce anemia. Moreover, strengthening women's education and policies regulating the prohibition of polygamous marriage is important to address the operational constraints. Our finding also suggests the need for further studies on the effectiveness of context-specific anemia interventions for this age group.

## Acknowledgments

The authors thank ICF International for granting access to the dataset used in this study.

## Author Contributions

**Conceptualization:** Kusse Urmale Mare, Setognal Birara Aychiluhm, Kebede Gemeda Sabo, Abay Woday Tadesse, Bizunesh Fentahun Kase, Oumer Abdulkadir Ebrahim, Tsion Mulat Tebeje, Getahun Fentaw Mulaw, Beminate Lemma Seifu.

**Data curation:** Kusse Urmale Mare, Getahun Fentaw Mulaw, Beminate Lemma Seifu.

**Formal analysis:** Kusse Urmale Mare, Tsion Mulat Tebeje, Beminate Lemma Seifu.

**Methodology:** Kusse Urmale Mare, Setognal Birara Aychiluhm, Kebede Gemeda Sabo, Abay Woday Tadesse, Bizunesh Fentahun Kase, Oumer Abdulkadir Ebrahim, Tsion Mulat Tebeje, Getahun Fentaw Mulaw.

**Software:** Kusse Urmale Mare, Beminate Lemma Seifu.

**Validation:** Kusse Urmale Mare, Beminate Lemma Seifu.

**Visualization:** Kusse Urmale Mare.

**Writing – original draft:** Kusse Urmale Mare, Setognal Birara Aychiluhm, Kebede Gemeda Sabo, Abay Woday Tadesse, Bizunesh Fentahun Kase, Oumer Abdulkadir Ebrahim, Tsion Mulat Tebeje, Getahun Fentaw Mulaw, Beminate Lemma Seifu.

**Writing – review & editing:** Kusse Urmale Mare, Setognal Birara Aychiluhm, Kebede Gemeda Sabo, Abay Woday Tadesse, Bizunesh Fentahun Kase, Oumer Abdulkadir Ebrahim, Tsion Mulat Tebeje, Getahun Fentaw Mulaw, Beminate Lemma Seifu.

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
