## [Decision Letter · Decision Letter 0]

23 Aug 2023

PONE-D-23-04658

Determinants of anemia level among reproductive-age women in Sub-Saharan African countries: a multilevel mixed-effects analysis with ordered logistic regression modeling

PLOS ONE

Dear Dr. Mare,

Thank you for submitting your manuscript to PLOS ONE. After careful consideration, we feel that it has merit but does not fully meet PLOS ONE’s publication criteria as it currently stands. Therefore, we invite you to submit a revised version of the manuscript that addresses the points raised during the review process.

Please submit your revised manuscript by Oct 07 2023 11:59PM. If you will need more time than this to complete your revisions, please reply to this message or contact the journal office at plosone@plos.org. Please include the following items when submitting your revised manuscript:

We look forward to receiving your revised manuscript.

Kind regards,

Fekede Asefa Kumsa, PhD

Academic Editor

PLOS ONE

Journal Requirements:

Additional Editor Comments:

There are numerous studies on anemia among similar population in sub-Saharan Africa, utilizing DHS data. Please explain what distinguishes your analysis from the existing studies. Additionally, ensure that you accurately interpret your findings.

Reviewers' comments:

Reviewer's Responses to Questions

**Comments to the Author**

1. Is the manuscript technically sound, and do the data support the conclusions?

Reviewer #1: Yes

Reviewer #2: Yes

Reviewer #3: Yes

2. Has the statistical analysis been performed appropriately and rigorously? 

Reviewer #1: Yes

Reviewer #2: Yes

Reviewer #3: No

3. Have the authors made all data underlying the findings in their manuscript fully available?

Reviewer #1: Yes

Reviewer #2: Yes

Reviewer #3: Yes

4. Is the manuscript presented in an intelligible fashion and written in standard English?

Reviewer #1: Yes

Reviewer #2: Yes

Reviewer #3: Yes

5. Review Comments to the Author

Reviewer #1: The paper is well written and the findings and conclusions are supported by the data. The research problem of the study is clearly defined and easily understandable and the objective of the paper has been well answered and addressed. I propose the paper be accepted for publication.

Reviewer #2: The authors have addressed an important public health concerns in SSA

I would like the authors to respond to these issues I have raised

1. 106 data that was collected earlier than 2010 (Central African Republic, Sudan, Nigeria Ondo State,

'Nigeria Ondo State' is not a country. Ondo State is one of the 37 state administrations in Nigeria. Please find out what DHS mean by Nigeria Ondo State and why it was conducted in isolation from the NDHS.

2. 113 and then appended them after managing missing observations.

Please, report how you managed the missing observations.

3. 140 First, bivariable 141 ordinal logistic regression analysis was done and proportional odds assumption was checked for 142 each explanatory variable using a post-estimation test (i.e. Brant test) [35, 36].

This procedure you have carried out is not very clear. Do you mean unadjusted ordinal logistic regression was done and Brant test was conducted for each predictor variables as post-estimation? Please make it clearer

Secondly, I am not sure if it is a gold standard to conduct brant test for single level would mean satisfying the proportional odds assumptions. The references you added conducted the Brant test for single level, and not multilevel. To avoid misleading your reading audience seeing this practice as the gold and acceptable standard, you may wish to include in your limitations on what you have done.

4. 204 final multilevel ordinal logistic regression model. It was revealed that women in a polygamous 205 union [AOR (95% CI) = 1.16(1.12, 1.21)] had a 16% greater chance of having a higher anemia 206 level than those in a monogamous union.

I have noticed that you have interpreted the results of ML ordinal logistic results as it would have been if they were binary cases.

I think the interpretation should be something like this:

“It was revealed that women in a polygamous union [AOR (95% CI) = 1.16(1.12, 1.21)] had a 16% greater chance of having a higher anemia versus the combined of no anemia, mild anemia, and moderate anemia, level when compared with those in a monogamous union”.

5. 216 underweight [AOR (95% CI) = 1.21(1.10, 1.27)], had unimproved toilet [AOR (95% CI) = 1.20 217 (1.16, 1.24)], unimproved water source [AOR (95% CI) = 1.34(1.30, 1.38)],

In DHS data, wealth status was created using some variables as proxies which include improved toilet, water source, etc. In your multivariate analysis, I am not expecting you to include these variables (proxies of wealth status) simultaneously with the wealth status. Do you think it is the right thing to do?

Reviewer #3: The manuscript sounds very interesting and the promising data can significantly add information on the anemia level and influencing factors in SSA. However, some parts are very hard to read, and as well as there are a consistent number of statistical issues that need to be considered.

6. PLOS authors have the option to publish the peer review history of their article (what does this mean?). If published, this will include your full peer review and any attached files.

Reviewer #1: No

Reviewer #2: **Yes: **Phillips Obasohan

Reviewer #3: No

---

## [Author Response · Author response to Decision Letter 0]

4 Sep 2023

Thank you for the opportunity to revise our manuscript entitled “Determinants of anemia level among reproductive-age women in 29 Sub-Saharan African countries: a multilevel mixed-effects modeling with ordered logistic regression analysis (ID: PONE-D-23-04658)”. We have intensively addressed the concerns raised by the reviewers using a point-by-point response as stated below. The amendments made to the manuscript have been presented using track change in the attachment titled “Revised manuscript with track changes”.

Responses to Editor’s Comments 

Comment: There are numerous studies on anemia among similar population in sub-Saharan Africa, utilizing DHS data. Please explain what distinguishes your analysis from the existing studies. Additionally, ensure that you accurately interpret your findings.

Response: Thank you for your important concern. As stated in the revised manuscript, we have indicated the existing gap in addition to methodological gaps as follows. 

“Evidence on the level of anemia and its determinants at the SSA level is limited and most of the previous studies across this region were limited to a single country [24, 30-33] and some African countries [20, 26, 34, 35]. We have thoroughly checked and corrected the interpretation of the results.

Responses to Reviewer-1 Comments 

Comment: The paper is well written and the findings and conclusions are supported by the data. The research problem of the study is clearly defined and easily understandable and the objective of the paper has been well answered and addressed. I propose the paper be accepted for publication.

Response: Thank you very much for accepting our manuscript in its current form.

 

Responses to Reviewer-2 Comments 

The authors have addressed an important public health concerns in SSA. I would like the authors to respond to these issues I have raised.

Comment 1: 106 data that was collected earlier than 2010 (Central African Republic, Sudan, Nigeria Ondo State, 'Nigeria Ondo State' is not a country. Ondo State is one of the 37 state administrations in Nigeria. Please find out what DHS mean by Nigeria Ondo State and why it was conducted in isolation from the NDHS.

Response 1: Thank you for your important concern. We agree with your concern that Nigeria Ondo State is one of state administration in Nigeria, however in the DHS database; the dataset is available for both Nigeria and Nigeria Ondo State separately. Most importantly, data from Nigeria Ondo State was not used in the current analysis, since the survey was conducted earlier than 2010.

Comment 2: 113 and then appended them after managing missing observations. Please, report how you managed the missing observations.

Response 2: Thank you for your suggestion. We have explained this concern in the revised manuscript.

Comment 3: 140 First, bivariable 141 ordinal logistic regression analysis was done and proportional odds assumption was checked for 142 each explanatory variable using a post-estimation test (i.e. Brant test) [35, 36]. This procedure you have carried out is not very clear. Do you mean unadjusted ordinal logistic regression was done and Brant test was conducted for each predictor variables as post-estimation? Please make it clearer.

Response 3: Thank you for your concern. As you said, it is to mean, “Unadjusted ordinal logistic regression was done and Brant test was conducted for each predictor variable as post-estimation” and we have corrected it as suggested in the revised manuscript.

Comment 4: Secondly, I am not sure if it is a gold standard to conduct Brant test for single level would mean satisfying the proportional odds assumptions. The references you added conducted the Brant test for single level, and not multilevel. To avoid misleading your reading audience seeing this practice as the gold and acceptable standard, you may wish to include in your limitations on what you have done.

Response 4: Thank you for your concern. As stated, this test is descriptive and used to check whether each explanatory variable has fulfilled the proportional odds assumption i.e. it can be used with ordinal outcome variable regardless of the data structure (single level/hierarchical) since it is descriptive.

Comment 5: 204 final multilevel ordinal logistic regression model. It was revealed that women in a polygamous 205 union [AOR (95% CI) = 1.16(1.12, 1.21)] had a 16% greater chance of having a higher anemia 206 level than those in a monogamous union.

I have noticed that you have interpreted the results of ML ordinal logistic results as it would have been if they were binary cases. I think the interpretation should be something like this:

“It was revealed that women in a polygamous union [AOR (95% CI) = 1.16(1.12, 1.21)] had a 16% greater chance of having a higher anemia versus the combined of no anemia, mild anemia, and moderate anemia, level when compared with those in a monogamous union”.

Response 5: Thank you for your suggestion. We have incorporated the suggested interpretation in the revised manuscript. 

Comment 6: 216 underweight [AOR (95% CI) = 1.21(1.10, 1.27)], had unimproved toilet [AOR (95% CI) = 1.20 217 (1.16, 1.24)], unimproved water source [AOR (95% CI) = 1.34(1.30, 1.38)],

In DHS data, wealth status was created using some variables as proxies which include improved toilet, water source, etc. In your multivariate analysis, I am not expecting you to include these variables (proxies of wealth status) simultaneously with the wealth status. Do you think it is the right thing to do?

Response 6: Thank you very much. We agree with your concern that these variables may be correlated. However, the selection of variables in the current study was based on the literature and the existence of multicollinearity. Most of the previous studies reported the effect of these variables (toilet facility, water source, and household wealth) on anemia, and on collinearity diagnostic, there was no evidence of multicollinearity. Due to this, we considered them in the final analysis.

Response to Reviewer -3 Comments 

I thank you for the opportunity to read and revise the manuscript “Determinants of anemia level among reproductive-age women in Sub-Saharan African countries: a multilevel mixed-effects analysis with ordered logistic regression modeling”.

The manuscript sounds very interesting and the promising data can significantly add information on the anemia level and influencing factors in SSA. However, some parts are very hard to read, and as well as there are a consistent number of statistical issues that need to be considered.

Requested revisions:

Abstract 

Comment 1: In method section “Data from 205,627 reproductive-age women from the recent demographic and health survey of ….” Which DHS conducted year you used please specify.

Response 1: Thank you very much. We have specified the survey year as suggested in the revised manuscript.

Comment 2: Methods used should be reflected in the abstract very well and precisely.

Response 2: Thank you. We have provided the details of the methods used as suggested in the revised manuscript.

Comment 3: Result part needs re-write and revision.

Response 3: Thank you. We have revised it accordingly based on the study’s objectives.

Introduction

Comment 1: Clear gap should be present other than methodological gap.

Response 1: Thank you for your important concern. As stated in the revised manuscript, we have indicated the existing gap in addition to methodological gaps as follows. 

“Evidence on the level of anemia and its determinants at the SSA level is limited and most of the previous studies across this region were restricted to a single country [24, 30-33] and some African countries [20, 26, 34, 35]. 

Comment 2: In general the background needs extensive revision.

Response 2: Thank you for your suggestion. We have revised this section accordingly.

Methods 

Comment 1: Please use sub-heading as “data extraction method”, I haven’t see data extraction method and where it was obtained.

Response 1: Thank you very much for your concern. We have provided the information regarding data extraction under the sub-heading “Data extraction and management of missing observations”. In addition, details about where data was obtained are indicated under the sub-heading “Data source”.

Comment 2: Line 113 “…. After managing missing observations” which missing mechanism technique you apply in this study. Please use “missing mechanism “as sub-heading explains more how you handle you missing data in your study.

Response 2: Thank you for your concern. As stated under the “Data extraction and management of missing observations” sub-heading, this is to indicate that women who had a missing observation on the outcome variable were dropped/excluded from the analysis. 

Comment 3: Data cleaning procedure was not mention

Response 3: Thank you very much. Information regarding data cleaning was stated under the “Data management and statistical analysis” section of the revised manuscript.

Comment 4: For the whole method section, please try to reflect the real activities undertaken by organizations who conducted the method and data collection.

Response 4: Thank you. As mentioned in the method section, we have used secondary data from DHS of 29 Sub-Saharan African countries, which was collected through the collaboration of the Central Statistics Agency of each country and ICF international. Details about activities undertaken by organizations can be accessed online at: 

(https://dhsprogram.com/Methodology/index.cfm).

Comment 5: The other serious problem, that requires serious attention, I have seen in this section is that, the authors did fall short of accounting for the complexity of the DHS data. The data were collected via multi-stage sampling approach with clustering and stratification. So, this data should be analyzed with this sampling process in mind. See the DHS analysis manual for better understanding. I would argue that findings without accounting for the complexity of the data are erroneous and biased towards oversampled areas and settings, and would not reflect the situation of each country.

Response 5: Thank you very much for your important concern. As stated in the methods section of the manuscript, we have accounted for the complexity of the sampling design by applying a multilevel modeling with an ordinal logistic regression analysis.

Comment 6: For each category of outcome variable cut point was not mention i.e. for not anemic, mild anemic, moderate anemic and sever anemic and cite where the cut point you found.

Response 6: Thank you very much for your interesting comment. We have mentioned the cut-off point for each anemia level and supported it with citations as suggested.

Comment 7: I haven’t seen any inclusion and exclusion criteria for this study.

Response 7: Thank you. We have excluded women who had a missing observation on the outcome variable (anemia level) from the study.

Data management and statistical analysis

Comment 1: How you judge the variable to have a statistically significant association with anemia level? And how you link the strength ….. Please write in this section.

Response 1: Thank you very much. Information about the statistical significance of the independent variables is stated under “Model building and selection” of the revised manuscript.

Comment 2: For the logistic regression model, you are going to use the Generalized Variance Inflation Factor (GVIF) since this model belongs to the class of generalized linear models. Results on GVIF are not presented.

Response 2: Thank you very much for an interesting suggestion. We have computed GVIF for all variables included in the final model and presented the result in Table 1.

Model building and selection

Comment 1: The model parameter estimation technique was not mentioned

Response 1: Thank you very much. The estimation of model parameters is stated under the “Model building and selection” of the revised manuscript.

Comment 2: What is your alternative model if the proportional odds model is failed…. I haven’t seen any other alternative ordinal model if the proportional odds model is failed….. I recommend the researcher to write other ordinal models (i.e. generalized ordered models, partial proportional odds model …) and when it applies these models in the method section.

Response 2: Thank you. We agree with your recommendation that other ordinal models should be considered when the proportional odds assumption is violated. However, as stated in the “Data management and statistical analysis” section of the revised manuscript, this assumption was satisfied in our analysis. Thus, we did not consider/discuss other models since the proportional odds assumption was fulfilled.

Comment 3: Is deviance the model selection criteria??? The model selection criteria are based on AIC and BIC values but I haven’t seen in this section. In addition testing overall model fit is not mention.

Response 3: Thank you very much. Deviance value (-2 * (Log Likelihood) is used as a model comparison since used a nested model (a multilevel). We have also presented other model section parameters (AIC, BIC, and LL) in Table 4 of the revised manuscript.

Comment 4 recommend including “marginal effect” i.e. the effect of each response category across the country.

Response 4: Thank you for your recommendation. The effect of each variable was presented in the regression table.

 

Results

Comment 1: I think prevalence of anemia among reproductive-age woman should present for each country.

Response 1: Thank you. We have presented the prevalence of anemia for each country in Figure-1. 

Comment 2: Result interpretation needs revision.

Response 2: Thank you very much. We have revised this section as suggested.

Comment 3: I think for this study country variation (substantial heterogeneity across each country) should reflect other than pooled prevalence.

Response 3: Thank you for your recommendation. We have reflected the variation in the anemia level across the included countries in addition to the pooled prevalence in the revised manuscript. In addition, details about the anemia level for each country are presented in Figure-1.

Discussion

Comment 1: Some parts in the discussion are difficult to understand. Please carefully rewrite this section considering every single set of covariates reporting both significant and non-significant results.

Response 1: Thank you for your suggestion. We have revised this section accordingly. 

Comment 2: Go through the whole document and edit technical issues

Response 2: Thank you. We have edited the manuscript to address grammar and editorial errors.

Conclusion

Comment 1: Your conclusion should be revised and Please add the suggestion based on the result and for the further studies.

Response 1: Thank you for your important recommendation. We have revised this section and provided suggestions for further studies.

---

## [Decision Letter · Decision Letter 1]

16 Oct 2023

PONE-D-23-04658R1Determinants of anemia level among reproductive-age women in 29 Sub-Saharan African countries: a multilevel mixed-effects modeling with ordered logistic regression analysisPLOS ONE

Dear Dr. Mare,

Thank you for submitting your manuscript to PLOS ONE. After careful consideration, we feel that it has merit but does not fully meet PLOS ONE’s publication criteria as it currently stands. Therefore, we invite you to submit a revised version of the manuscript that addresses the points raised during the review process.Please also find additional comments made by the reviewer 2, which needs to be addressed before accepting the paper for publication. 

We look forward to receiving your revised manuscript.

Kind regards,

Fekede Asefa Kumsa, PhD

Academic Editor

PLOS ONE

Journal Requirements:

Reviewers' comments:

Reviewer's Responses to Questions

**Comments to the Author**

1. If the authors have adequately addressed your comments raised in a previous round of review and you feel that this manuscript is now acceptable for publication, you may indicate that here to bypass the “Comments to the Author” section, enter your conflict of interest statement in the “Confidential to Editor” section, and submit your "Accept" recommendation.

Reviewer #2: (No Response)

2. Is the manuscript technically sound, and do the data support the conclusions?

Reviewer #2: Yes

3. Has the statistical analysis been performed appropriately and rigorously? 

Reviewer #2: Yes

4. Have the authors made all data underlying the findings in their manuscript fully available?

Reviewer #2: Yes

5. Is the manuscript presented in an intelligible fashion and written in standard English?

Reviewer #2: Yes

6. Review Comments to the Author

Reviewer #2: I am afraid, you have not advanced enough reasons for the following comments

Comment 1. The Nigeria Ondo State survey is not needed in this analysis, therefore it should be presented in such a manner that it will not confuse the readers to assume that the Ondo State is one among the national surveys. If it is not needed, then remove it from your statements. It might be mis-leading others who may want to include data beyond your 2010 limit, and may assume Nigeria Ondo State is a national representative survey.

Comment 4. I am not satisfied with the reasons given for using the Brant test to evaluate proportionality assumptions for an ML. It is not a gold standard. The fact that others may have assumed it, as far as I know, is not the gold standard. A statement of caution should be made either at the point of usage or as a limitation. The danger is that when your paper is published as it is other researchers will begin to reference your paper as standard. If you found references that have substantiated the validity of using the Brant test for ML, you may include it in your paper

Comment 5. You did not change anything in the interpretations as you have stated in your comment. I can't find any change.

Comment 6. I do not agree that you can add simultaneously variables that you have used as proxies to 'wealth status' in a multivariate model. If it is in an unadjusted model, that will be understood. Simply that the multicollinearity check could not fish it out, will not make it the right thing to do. This may affect the reproducibility of your work. If you have references that have proved that this procedure is acceptable, then include them in your paper, otherwise, let it be clear to your reader what you have done, and why you decided to do so.

7. PLOS authors have the option to publish the peer review history of their article (what does this mean?). If published, this will include your full peer review and any attached files.

Reviewer #2: **Yes: **Phillips Obasohan

---

## [Author Response · Author response to Decision Letter 1]

13 Nov 2023

Thank you for the opportunity to revise our manuscript entitled “Determinants of anemia level among reproductive-age women in 29 Sub-Saharan African countries: a multilevel mixed-effects modelling with ordered logistic regression analysis (ID: PONE-D-23-04658R1)”. We have intensively addressed the concerns raised by the reviewers using a point-by-point response as stated below. The amendments made to the manuscript have been presented using track change in the attachment titled “Revised manuscript with track changes”.

Responses to Editor’s- Comments 

Response: Thank you very much. We have checked all references for the stated concern and confirm that there was no retracted reference in the list. 

Responses to Reviewer-2 Comments 

I am afraid, you have not advanced enough reasons for the following comments:

Comment 1. The Nigeria Ondo State survey is not needed in this analysis, therefore it should be presented in such a manner that it will not confuse the readers to assume that the Ondo State is one among the national surveys. If it is not needed, then remove it from your statements. It might be mis-leading others who may want to include data beyond your 2010 limit, and may assume Nigeria Ondo State is a national representative survey.

Response: Thank you for your recommendation. We have removed Nigeria Ondo State as suggested from the revised manuscript.

Comment 4. I am not satisfied with the reasons given for using the Brant test to evaluate proportionality assumptions for an ML. It is not a gold standard. The fact that others may have assumed it, as far as I know, is not the gold standard. A statement of caution should be made either at the point of usage or as a limitation. The danger is that when your paper is published as it is other researchers will begin to reference your paper as standard. If you found references that have substantiated the validity of using the Brant test for ML, you may include it in your paper.

Response: Thank you again for your important concern. As per the literarture, there are many tests that are used to assess the proportionality assumption in ordinal logistic regression analyses like Wald, likelihood ratio (LR), Brant, and Wolfe and Gould. As stated previously these tests are descriptive and used to check whether each explanatory variable has fulfilled the proportional odds assumption i.e. it can be used with ordinal outcome variable regardless of the data structure (single level/hierarchical). We have rechecked the assumption using likelihood ratio (LR) and Brant tests. However, we were unbale to run the other tests since there is no STATA command for the remaining tests. Thus we have acknowledged this concern in the limitation section as suggested.

Comment 5. You did not change anything in the interpretations as you have stated in your comment. I can't find any change.

Response: Thank you very much for your rimnder and sorry for the technical error. We have changed the interpretation as previously suggested in the revised manuscript.

Comment 6. I do not agree that you can add simultaneously variables that you have used as proxies to 'wealth status' in a multivariate model. If it is in an unadjusted model, that will be understood. Simply that the multicollinearity check could not fish it out, will not make it the right thing to do. This may affect the reproducibility of your work. If you have references that have proved that this procedure is acceptable, then include them in your paper, otherwise, let it be clear to your reader what you have done, and why you decided to do so.

Response: Thank you once again. As we stated in the previous reposne, we agree with your concern that these variables may be correlated. However, the selection of variables in the current study was based on the literature and the existence of multicollinearity. Previous studies have reported the effect of these variables (toilet facility, water source, and household wealth) on anemia. Moreover on the collinearity diagnostic, there was no evidence of multicollinearity, due to this, we considered them in the final analysis. Please, see the articles that have reported these vaiables below: 

https://doi.org/10.1186/s13690-021-00733-x

https://doi.org/10.1016/j.cegh.2021.100948

https://doi.org/10.1371/journal.pone.0236449

https://doi.org/10.3390/nu13082745

---

## [Editor Report · Decision Letter 2]

14 Nov 2023

Determinants of anemia level among reproductive-age women in 29 Sub-Saharan African countries: a multilevel mixed-effects modelling with ordered logistic regression analysis

PONE-D-23-04658R2

Dear Dr. Mare,

We’re pleased to inform you that your manuscript has been judged scientifically suitable for publication and will be formally accepted for publication once it meets all outstanding technical requirements.

Kind regards,

Fekede Asefa Kumsa, PhD

Academic Editor

PLOS ONE

---

## [Editor Report · Acceptance letter]

16 Nov 2023

PONE-D-23-04658R2 

Determinants of anemia level among reproductive-age women in 29 Sub-Saharan African countries: a multilevel mixed-effects modelling with ordered logistic regression analysis 

Dear Dr. Mare:

I'm pleased to inform you that your manuscript has been deemed suitable for publication in PLOS ONE. Congratulations! Your manuscript is now with our production department. 

Kind regards, 

on behalf of

Dr. Fekede Asefa Kumsa 

Academic Editor

PLOS ONE